# Lineax: unified linear solves and linear least-squares in JAX and Equinox

**Jason Rader**
Oxford University
rader@maths.ox.ac.uk

**Terry Lyons**
Oxford University

**Patrick Kidger**
Google X
math@kidger.site

## Abstract

We introduce Lineax, a library bringing linear solves and linear least-squares to the JAX+Equinox scientific computing ecosystem. Lineax uses general linear operators, and unifies linear solves and least-squares into a single, autodifferentiable API. Solvers and operators are user-extensible, without requiring the user to implement any custom derivative rules to get differentiability. Lineax is available at https://github.com/google/lineax.

## 1 Introduction

JAX is an autodifferentiatiable Python framework popular for machine learning and scientific computing (4; 9; 12; 16). Equinox (20) is a popular JAX library (8; 15), targeting the same use cases, that adds additional support for parameterised functions. Solving linear systems, whether well-posed linear solves or ill-posed linear least-squares problems, is a central sub-problem in scientific computing (14; 27). For example, linear solves and least-squares appear as subroutines in nonlinear optimisation (21), finite-difference schemes (26), and signal processing (22). As such, we introduce Lineax, a library built in JAX and Equinox for linear solves and linear least-squares.

Lineax presents a single, differentiable interface for solving well-posed, underdetermined, and overdetermined linear systems. It also allows users to write custom differentiable linear solvers or least-squares solvers, and introduces a linear operator abstraction.

Overall, we intend for Lineax to integrate well with the existing JAX scientific ecosystem. This ecosystem is growing, and includes packages for differentiable rigid-body physics simulation (11), computational fluid dynamics (3; 6), protein structure prediction (10), ordinary and stochastic differential equations (19), and probabilistic modeling (25). We are beginning to see some use of Lineax in this ecosystem already. This includes for linear subroutines in ocean dynamics (18) and optimal transport (5). Further, Diffrax (19) plans to adopt Lineax in the near future for linear subroutines in differential equations solves.

### 1.1 Main contributions

The main contributions of Lineax are:

- A general linear operator abstraction, as implemented by dense matrices, linear functions, Jacobians, etc.
- Stable and fast gradients through least-squares solves. This includes through user-defined solvers, without requiring extra effort from the user.
- PyTree-valued [1] operators and vectors.

---

[1] JAX terminology for arbitrarily nested container 'node' types (tuples/dictionaries/lists/custom types) containing 'leaves' (every other Python/JAX type.) We exclusively consider PyTrees who's leaves are JAX arrays.

NeurIPS 2023 AI for Science Workshop.

**Comparisons to existing JAX APIs**

The operator abstraction introduced in Lineax offers a flexibility not found in core JAX, which supports only dense matrices or matrix-vector product representations of operators. Lineax introduces new solvers over core JAX, such as `lineax.Tridiagonal`. Lineax also offers a consistent API between operators and solvers, which is what allows for extensibility to user-specified custom operator and solvers.

Compilation times for most Lineax solvers are essentially identical to JAX native solvers; Lineax's iterative solvers (CG, GMRES, ...) compile roughly twice as fast. The 'benchmarks' folder on GitHub provides a quantitative comparison.

We emphasise the stable and fast gradients to contrast with the existing JAX implementation, which as of version 0.4.16 exhibits instability or incorrect gradients in some exceptional cases.

For these reasons, JAX is actually considering deprecating some of its own APIs in favour of Lineax (29).

## 1.2 Classical linear solve example

Consider solving $Ax = b$ for a random matrix $A \in \mathbb{R}^{10 \times 10}$ against a random vector $b \in \mathbb{R}^{10}$. This can be done via

```
1    import jax.random as jr
2    import lineax as lx
3
4    A_key, b_key = jr.split(jr.PRNGKey(0))
5    A = lx.MatrixLinearOperator(jr.normal(A_key, (10, 10)))
6    b = jr.normal(b_key, (10,))
7    solution = lx.linear_solve(A, b)
```

# 2 Performing linear solves and least-squares

The main entry point to linear solves and least-squares in Lineax is

```
     lineax.linear_solve(A, b, solver)
```

for a linear operator $A$ and PyTree $b$. This performs a linear solve $Ax = b$ (for well-posed systems), or returns a least-squares solution $\min_x \|Ax - b\|_2$ (for overdetermined systems), or returns a minimum norm solution $\min_x \|x\|_2$ subject to $Ax = b$ (for underdetermined systems). This is a lot of operations to unify together, and it may initially seem strange to do so. The common thread – and our justification for unifying these operations – is that mathematically, all the above operations correspond to the pseudoinverse solution to $Ax = b$, ie. the solution arising from using the Moore-Penrose pseudoinverse $x = A^\dagger b$ (1; 23).

The user can specify which solver they'd like to use via the `solver` argument. This is helpful when the user already knows which solvers should work well for a problem. Not every solver is capable of handling every problem. For example, `lineax.CG` handles positive definite operators (21, section 5). Using a solver with an incompatible problem will result in an error.

# 3 General linear operators

In Lineax, we represent $A$ more generally than as an $n \times m$ matrix. Instead, we represent $A$ as a linear operator $A : X \to Y$, where $X$ and $Y$ are spaces of PyTrees of arrays. At an implementation level, a linear operator is an object which subclasses

```
     lineax.AbstractLinearOperator.
```

When $A$ is a dense matrix, $A \in \mathbb{R}^{\dim(Y) \times \dim(X)}$, it can be treated as a Lineax linear operator via

```
     lineax.MatrixLinearOperator(A).
```

Lineax operators themselves form a vector space, and are closed under addition, scalar multiplication, and composition. Each linear operator $A$ must implement a method to:

- Compute the matrix-vector product: $Ax$ for $x \in X$.
- Compute the transpose of the operator: $A^T : Y \to X$.
- Materialise the operator as a matrix: $A$.as_matrix() $\in \mathbb{R}^{\dim(X) \times \dim(Y)}$.
- Retrieve the input/output PyTree structure, as well as the input/output dimensions. ie. the functions $\text{domain}(A) = X$ and $\text{codomain}(A) = Y$.

This increased generality comes with increased flexibility. For example: large, sparse matrices can use data-efficient formats and utilise linear solves which use only the matrix-vector product, such as GMRES (24) or BiCGStab (28). For example, a linear function $f : X \to Y$ can be made into a linear operator with

```
lineax.FunctionLinearOperator(f, in_structure)
```

where `in_structure` describes the PyTree structure of the input of $f$ (equivalently, the PyTree structure of the elements $x \in X$.) Similarly, a nonlinear function $g : X \to Y$ can be linearised at a point $x \in X$ and use its Jacobian at $x$ as a linear operator via

```
lineax.JacobianLinearOperator(g, x)
```

The `lineax.AbstractLinearOperator` base class is available for users to subclass and create their own linear operator types.

## 3.1 Operator tags

Tags are an optional argument to most linear operators, and indicate properties of the operator $A$. For example, if $A \in \mathbb{R}^{n \times n}$ is positive semidefinite, then $A$ can be marked as a positive semidefinite linear operator with

```
lineax.MatrixLinearOperator(A, lineax.positive_semidefinite_tag)
```

This indicates to any solver which uses $A$ that it is positive semidefinite. For example, if $A$ is also nonsingular, then it can be used safely with `lineax.CG`. Tags are also used to select the appropriate solver in the polyalgorithm `lineax.AutoLinearSolver` detailed in section 6.

## 4 Computing gradients

In JAX, derivatives are built from Jacobian-vector products (JVPs) and vector-Jacobian products (VJPs) for forward-mode and reverse-mode automatic differentiation respectively (12). The JVP of a function $f : \mathbb{R}^a \to \mathbb{R}^b$ maps an input-tangent pair $(x, v) \in \mathbb{R}^a \times \mathbb{R}^a$ to $(f(x), \partial f(x)(v)) \in \mathbb{R}^b \times \mathbb{R}^b$ where $\partial f(x) : \mathbb{R}^a \to \mathbb{R}^b$ is the Jacobian of $f$ at $x$. The VJP maps an input-cotangent pair $(x, c) \in \mathbb{R}^a \times \mathbb{R}^b$ to $(f(x), \partial f(x)^T c) \in \mathbb{R}^b \times \mathbb{R}^a$, where $\partial f(x)^T : \mathbb{R}^b \to \mathbb{R}^a$ is the transpose of $\partial f(x)$.

The major contribution of Lineax over existing linear solve and least-squares software is the efficient computation of JVPs for pseudoinverse solutions. That is, differentiation through both well-posed linear solves and ill-posed least-squares solves are performed in the same manner as each other. In particular, we may special case when operators have full row or column rank in order to obtain improved performance, as we now show.

### 4.1 JVPs and forward-mode autodifferentiation

In this section, let $\mathcal{L}(A, b)$ denote the linear solve `lineax.linear_solve`. For a primal problem $Ax = b$, then $\mathcal{L}(A, b) = A^\dagger b$ where $A^\dagger$ be the Moore–Penrose pseudoinverse of $A$, as mentioned in section 2. Here we discuss how the to compute the JVP $\partial \mathcal{L}(A, b)(V, v)$, where $(V, v)$ is the tangent pair consisting of a tangent operator $V$ and tangent vector $v$.

It is possible to compute the JVP through either argument. For example, the tangent computation of a linear solve as a function of $v$ alone is

$$\partial \mathcal{L}(A, b)(0, v) = A^\dagger v$$

where $0$ represents the $0$ tangent operator.

Meanwhile, computing the JVP for $\partial\mathcal{L}(A,b)(V,0)$ requires differentiating through a pseudoinverse, which has the explicit formula (13)

$$\partial\mathcal{L}(A,b)(V,0) = (-A^\dagger V A^\dagger + A^\dagger(A^\dagger)^T V^T(I - AA^\dagger) + (I - A^\dagger A)V^T(A^\dagger)^T A^\dagger)A^\dagger b.$$

Letting

$$x = A^\dagger b$$
$$z = V^T(A^\dagger)^T x,$$

and adding the above two equations together and using the linearity of the Jacobian, we have the total JVP with respect the primal pair $(A,b)$ and tangent pair $(V,v)$ for `lineax.linear_solve` is

$$\partial\mathcal{L}(A,b)(V,v) = A^\dagger\left(-Vx + (A^\dagger)^T V^T(b - Ax) - Az + v\right) + z. \tag{1}$$

If $A$ has linearly independent columns, then $A^\dagger A = I$ (14, section 5.5.2) and the term $z - A^\dagger Az = 0$, giving

$$\partial\mathcal{L}(A,b)(V,v) = A^\dagger\left(-Vx + (A^\dagger)^T V^T(b - Ax) + v\right). \tag{2}$$

When $A$ has linearly independent rows, then $(b - Ax) = 0$ and

$$\partial\mathcal{L}(A,b)(V,v) = A^\dagger\left(-Vx - Az + v\right) + z. \tag{3}$$

Together, if $A$ has linearly independent rows and columns, then $A$ is well-posed, $A^\dagger = A^{-1}$ is a true inverse, and

$$\partial\mathcal{L}(A,b)(V,v) = A^{-1}\left(-Vx + v\right). \tag{4}$$

We then select between equations (4), (3), (2), or (1) depending on whether we know at compile time that $A$ has linearly independent rows and columns, has only independent rows, has only independent columns, or has both dependent rows and columns. Despite being a property of the operator, at compile time the main way the JVP rule is dispatched via the choice of solver. This is because not every solver supports dependent rows/columns (section 5), and will return `nan` values if used in a solve with an unsupported operator. So, if a solver does not support dependent rows/columns, we can be sure we will not get a solution given an operator with dependent rows/columns in the JVP.

For example, `lineax.QR` (27, section 2) can handle dependent rows if the number of rows is greater than the number of columns (or dependent columns if the number of columns is greater than the number of rows) and will dispatch to either equation (2) or (3). If the number of columns and rows of $A$ are the same, then `lineax.QR` will dispatch to equation (4).

### 4.2 VJPs and backpropogation via transposition

Reverse-mode autodifferentiation of a function $f : \mathbb{R}^a \to \mathbb{R}^b$ is not built on JVPs, but rather on vector-Jacobian products (VJPs) $v^T\partial f(x) = \partial f(x)^T v$. As suggested by the definition, VJPs are constructed via a JVP and transposition (12). This is how VJPs are implemented in JAX, and thus in Lineax as well. The Jacobian $\partial\mathcal{L}(A,b) : \mathbb{R}^{m\times n} \times \mathbb{R}^m \to \mathbb{R}^n$ is a linear function, see also the explicit form in equation (1). Therefore, it has a transpose $\partial\mathcal{L}(A,b)^T : \mathbb{R}^n \to \mathbb{R}^{m\times n} \times \mathbb{R}^m$.

The transpose rule for the linear solve is implemented as a custom JAX primitive. See (12) for more details.

## 5 User-defined solvers

A user can implement a custom solver by subclassing

```
lineax.AbstractLinearSolver
```

which requires the methods: `init`, `compute`, `transpose`, `allow_dependent_rows`, and `allow_dependent_columns`.

Many direct linear solvers for $Ax = b$ use two stages of computation. First, factor $A$ into a form amenable to computation (eg. LU factorisation (14, section 3), QR factorisation, SVD factorisation, etc.) Then, use this factorisation to solve for a given right hand side $b$. The factorisation of $A$ does not depend on the right hand side $b$, and can be reused with various choices of $b$. This saves computation cost when solving $Ax = b$ for many right hands $b$.

For a solver using such a two-stage factorisation approach, `init` computes the factorisation, and `compute` performs the solve for the specific right hand $b$. `transpose` computes the transpose of the factorisation provided by `init`, which allows us to skip computing the factorisation of the transpose operator directly (`init(transpose(operator))`), as it is commonly the case that we can cheaply derive this from `init(operator)` alone. This is needed when computing VJPs as discussed in the previous section. The methods `allow_dependent_rows` and `allow_dependent_columns` determine which equation (1-4) is used in the differentiation, as discussed in section 4.

This greatly simplifies the process of writing a differentiable linear solver or least-squares solver. In the core JAX library, it is somewhat cumbersome to write a differentiable solver. It requires using `jax.lax.custom_linear_solve` and implementing a solver transposition rule `transpose_solve` if the user would like to use reverse-mode autodifferentiation. In Lineax, differentiation comes for free once a solver is implemented, whether the solver is a linear solve or a least-squares algorithm.

## 6 The `AutoLinearSolver` polyalgorithm

If the user does not provide a solver to `lineax.linear_solve`, then the default linear solve is `lineax.AutoLinearSolver`. `lineax.AutoLinearSolver` is a polyalgorithm which selects a solver automatically at compile time depending on the structure of $A$, as indicated through its operator tag (discussed in section 3.1). `lineax.AutoLinearSolver` takes the argument `well_posed`, which indicates whether the system is expected to solve a least-squares/minimum norm problem, or only handle well-posed linear solves.

`lineax.AutoLinearSolver(well_posed=True)` selects a solver depending upon the operator structure, and throws an error when it encounters an underdeteremined or overdetermined system. `lineax.AutoLinearSolver(well_posed=False)` solves well-posed linear solves as well as linear least squares, but often at an additional computation cost. Finally, `lineax.AutoLinearSolver(well_posed=None)` solves a least-squares problem only if it is not expensive to do so.

The specific polyalgorithms for `well_posed=True`, `well_posed=False`, and `well_posed=None` are shown in figure 1.

### 6.1 Choosing a solver at compile time

We must choose between two paradigms for the implementation of `lineax.AutoLinearSolver`: make the algorithm selection at run time, or make the algorithm selection at compile time. This is a trade-off, as determining which solver to use at run time means checking the elements of the matrix. This incurs a run time overhead of $\mathcal{O}(n^2)$ for an $n \times n$ matrix, which is relatively small compared to the $\mathcal{O}(n^3)$ run time of most linear solve algorithms. However, run time checking also incurs a greater cost in compile times. Since it is not known at compile time which branch of the polyalgorithm will run, the compiler is forced to compile all branches. Thus, compilation cost scales with the logic of the polyalgorithm: as more branches are included, compile times increase. This can limit the extensibility of the polyalgorithm.

Compile time selection avoids these performance issues, and is faster both in run time and compile time when used correctly. Further, it simplifies tying solves to GPU hardware, as there is no possibility of taking separate branches for different batch elements. However, compile time selection requires the user to a-priori know the structure of the operator, and can result in using a suboptimal solver if the operator has exploitable structure which the user does not indicate.

We choose the compile time approach, and require the user to pass the structure of an operator explicitly via the operator tag (section 3.1.) We choose this approach primarily to minimise compilation times, and to avoid the tradeoff between extensibility and compile time inherent in run time checking.

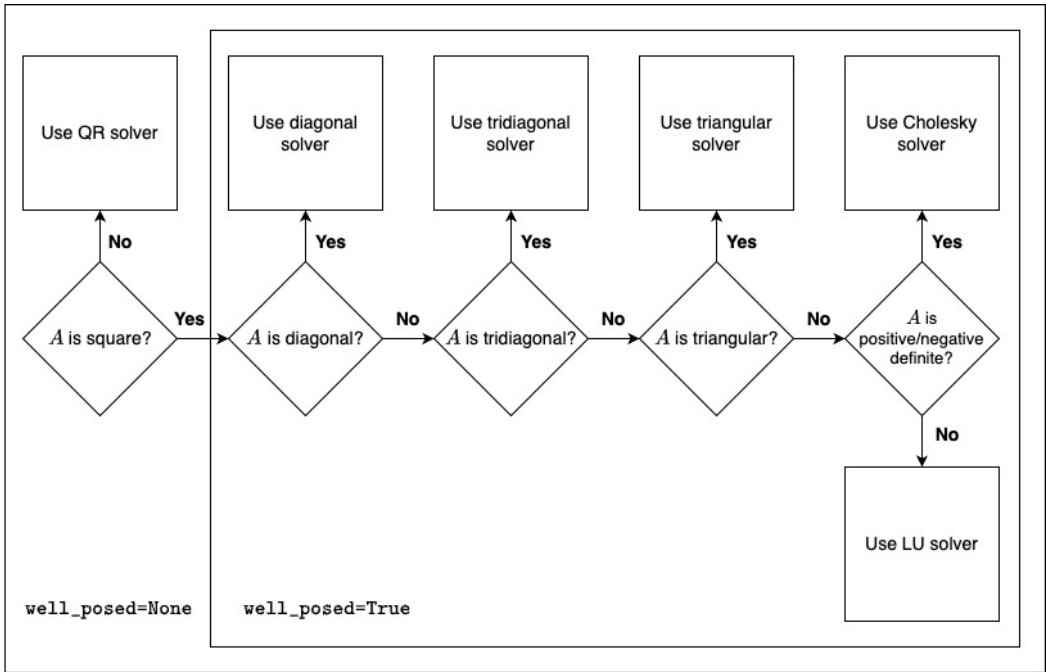

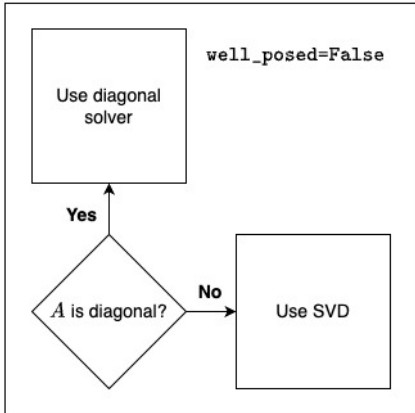

Figure 1: The `AutoLinearSolver` polyalgorithm. Read from left-to-right, so that `well_posed=True` starts at "$A$ is square?", `well_posed=None` starts at "$A$ is diagonal?", and `well_posed=False` starts at "$A$ is diagonal?" in the `well_posed=False` section.

Our approach is in contrast to MATLAB's `mldivide`, a (nondifferentiable) unified linear solve and least-squares solver which uses a run time approach (17). Both the Julia and MATLAB languages offer methods for nonsingular linear solves – the infix \ operation in Julia and `linsolve` in MATLAB – which accept compile time tags, but still perform run time checks if the user passes no tags (2; 7; 17). Therefore, both suffer from the additional overhead of the run time approach in many cases.

## 7   Conclusion

We have introduced Lineax, a differentiable JAX+Equinox library unifying linear solves and linear least-squares. We have demonstrated that users can extend base Lineax operators and solvers and use them within our unified API, without the need to write any custom derivative rules. We hope to see adoption of Lineax solves within the JAX+Equinox scientific computing and machine learning ecosystem.

## 8   Acknowledgements

This publication is based on work supported by the EPSRC Centre for Doctoral Training in Mathematics of Random Systems: Analysis, Modelling and Simulation (EP/S023925/1)

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
