# OpenReview forum: "Lineax: unified linear solves and linear least-squares in JAX and Equinox"
_NeurIPS.cc/2023/Workshop/AI4Science — NeurIPS2023-AI4Science Poster_

### Official Review · Reviewer_Voyd · 2023-10-25
**Authors have developed a solved called Lineax using JAX and Equinox to solve linear equation and linear-least square**

**Rating:** 7
**Confidence:** 3

**Review:**

Authors have developed a solved called Lineax using the existing framework of JAX and Equinox that is capable of solving linear  equation and linear-least square under different condition. The proposed library will be useful to other researcher as  many scientific problems requires solving linear equation such  as  in quantum simulation to simulate material property.  Thus this reviewer things this work will be good addition to the workshop
As an improvement, it will be great if author can also provide time complexity of  their library as a function of matrix size and how it compair with other linear solver  library such as BLAS

---

### Meta-Review · Area_Chair_UsT9 · 2023-10-26

**Recommendation:** Accept (Poster)
**Confidence:** 3

**Metareview:**

Reviewer agrees that the paper is well fit to the workshop scope and can potentially be useful in various scientific domains.